# Prediction of Dynamic Toxicity of Nanoparticles Using Machine Learning

**DOI:** 10.3390/toxics12100750

**Published:** 2024-10-15

**Authors:** Ivan Khokhlov, Leonid Legashev, Irina Bolodurina, Alexander Shukhman, Daniil Shoshin, Svetlana Kolesnik

**Affiliations:** 1Research Institute of Digital Intelligent Technologies, Orenburg State University, Pobedy Pr. 13, Orenburg 460018, Russia; iv.hohlov-01@yandex.ru (I.K.); ipbolodurina@yandex.ru (I.B.); shukhman@gmail.com (A.S.); svkolesnik_osu@mail.ru (S.K.); 2Federal Research Centre of Biological Systems and Agrotechnologies of the Russian Academy of Sciences, Orenburg 460000, Russia; daniilshoshin@mail.ru; 3Scientific and Educational Center “Biological Systems and Nanotechnologies”, Orenburg State University, Pobedy Pr. 13, Orenburg 460018, Russia

**Keywords:** nanoparticles, toxicity, prediction, artificial intelligence

## Abstract

Predicting the toxicity of nanoparticles plays an important role in biomedical nanotechnologies, in particular in the creation of new drugs. Safety analysis of nanoparticles can identify potentially harmful effects on living organisms and the environment. Advanced machine learning models are used to predict the toxicity of nanoparticles in a nutrient solution. In this article, we performed a comparative analysis of the current state of research in the field of nanoparticle toxicity analysis using machine learning methods; we trained a regression model for predicting the quantitative toxicity of nanoparticles depending on their concentration in the nutrient solution at a fixed point in time with the achieved metrics values of MSE = 2.19 and RMSE = 1.48; we trained a multi-class classification model for predicting the toxicity class of nanoparticles depending on their concentration in the nutrient solution at a fixed point in time with the achieved metrics values of Accuracy = 0.9756, Recall = 0.9623, F1-Score = 0.9640, and Log Loss = 0.1855. As a result of the analysis, we concluded the good predictive ability of the trained models. The optimal dosages for the nanoparticles under study were determined as follows: ZnO = 9.5 × 10^−5^ mg/mL; Fe_3_O_4_ = 0.1 mg/mL; SiO_2_ = 1 mg/mL. The most significant features of predictive models are the diameter of the nanoparticle and the nanoparticle concentration in the nutrient solution.

## 1. Introduction

Nanoparticles belong to the group of isolated ultrafine objects and are widely used in biomedical nanotechnologies [1], agricultural systems [2], microelectronics [3], the food industry [4], the manufacture of drugs [5], cosmetics [6], contrast agents for magnetic resonance imaging [7], energy generation [8], etc. Currently, scientists have identified and synthesized thousands of different nanoparticles varying in size, shape, composition, agglomeration, and homogeneity. An analysis of the toxicological effects of nanoparticles is necessary to identify their potential harmful effects on living organisms [9] and the environment [10]. It is worth noting that experimental assessment of the toxicity of various nanoparticles is an expensive and time-consuming procedure. Modern machine learning (ML) models can be used to classify the toxicity category and predict the viability of a nanoparticle based on in vitro and/or in vivo data from laboratory experiments. One of the main problems is the small size and/or insufficient initial data available for training models and the wide variety of nanoparticles that require careful analysis for toxicity. In this article, we carried out a study of two ML models to predict toxicity class and a dynamic toxicity quantifier depending on nanoparticle concentration in nutrient solution for SiO_2_, ZnO, and Fe_3_O_4_ nanoparticles.

A large number of current studies are devoted to the study of the toxicity of nanoparticles. S Medici et al. [11] review the origins, behavior, and biological toxicity of various metal nanoparticles to investigate health hazards in living organisms. M Horie et al. [12] examine the effect of oxidative stress on nanoparticle toxicity and conclude that particle size is not a direct factor influencing oxidative stress. Z Yu et al. [13] explore the unique physicochemical characteristics (such as ultra-small size, high surface area to mass ratio, and high reactivity) of nanoparticles and propose strategies to optimize nanoparticles for biomedical applications. C Egbuna et al. [14] review the toxic effects of nanoparticles and the factors underlying their toxicity, highlighting nanoparticle type, size, surface area, shape, aspect ratio, surface coverage, crystallinity, dissolution, and agglomeration. G. R. Tortella et al. [15] investigate the toxicity and antimicrobial properties of silver nanoparticles (AgNPs) and their effects on various aquatic and terrestrial organisms in the environment. AB Sengul et al. [16] examine the properties and toxicity of metal- and non-metal-based nanoparticles, noting the dependence of toxicity on the degree of oxidation, ligands, solubility, and morphology of the nanoparticles. A Sani et al. [17] evaluated the toxicological aspects of AuNPs in in vivo and in vitro experiments. L Vimercati et al. [18] perform an assessment of the potential acute toxicity of zinc nanoparticles (ZnO) on two species of marine crustaceans, noting that dissolution processes play a key role in the toxicity of ZnO. W Najahi-Missaoui et al. [19] note the importance of studying the toxicological status of known nanoparticles and optimizing their physicochemical properties while maximizing their biological effectiveness. T Jaswal et al. [20] are exploring the issues of minimizing the toxic effect of silver nanoparticles (AgNP) on human health through the use of the green synthesis approach. N Malhotra et al. [21] investigate the influence of physicochemical factors such as the hardness of the water, alkalinity, the presence of inorganic and organic ligands, pH levels, and temperature on the toxicity of copper nanoparticles to various fish species. R Ettlinger et al. [22] highlight chemical composition, dose, size, colloidality, and chemical stability in biological fluids as the most significant features affecting the toxicity of metal-organic framework nanoparticles. N Malhotra et al. [23] note in their study the high toxicity of magnetic iron oxide nanoparticles due to their large surface-to-volume ratio, chemical composition, size and dosage, retention in the body, immunogenicity, organ-specific toxicity, degradation, and excretion from the body. F Ameen et al. [24] investigate the toxicity of metal-based nanoparticles and their effect on various soil bacteria and fungi. R Abbasi et al. [25] note the direct influence of size, shape, structure, agglomeration state, surface characteristics, dose, type of material, responding cell type, and exposure time on the cytotoxicity of polymeric, silica-based, carbon-based, and metallic-based nanoparticles. SK Jha et al. [26] carry out a comprehensive study of the influence of physicochemical descriptors on the toxicity of nanoparticles using the principal component method. In total, the authors of the study identify 19 physicochemical descriptors based on five datasets and note that descriptors associated with the size of nanomaterials can be considered the most important indicators for cellular uptake and toxicity.

Some progress has been made in the field of quantitative prediction of the toxicity of nanoparticles based on physicochemical characteristics using modern machine learning methods. N Shirokii et al. [27] perform quantitative prediction of cytotoxicity of inorganic nanomaterials using the LightGBMRegressor ML model. I Furxhi et al. [28] perform classification prediction of the toxicity of nanoparticles based on a neural network (NN), random forest (RF), and voting as an ensemble meta-classifier to optimize the prediction results. S Kar et al. [29] perform classification prediction of the toxicity of metal oxide nanoparticles; linear discriminant analysis (LDA) shows the best results. F Zhang et al. [30] in their study present an effective strategy for predicting the toxicity of chemical mixtures of engineered nanoparticles (ENPs) based on NN and support vector machines (SVM). A Banaye Yazdipour et al. [31] review artificial intelligence tools for assessing the toxicity of nanomaterials and highlight RF and SVM as the most common and note their acceptable predictive accuracy results. G. P. Gakis et al. [32] use predictive models to measure nanoparticle toxicity to extract valuable mechanistic information. A Banerjee et al. [33] predict the cytotoxicity of TiO_2_-based multicomponent nanoparticles using the RF model. M Na et al. [34] performed nanoparticle toxicity analysis with four ML algorithms: RF, SVM, Bayesian regularized neural network (BRNN), and Multiple Linear Regression (MLR). AS Desai et al. [35] investigate the prediction of cytotoxicity in silver nanoparticles using RF and Decision Tree (DT). I Furxhi et al. [36] perform a binary classification of nanoparticle toxicity using RF and goodness-of-fit. Table 1 aggregates the results obtained in the mentioned studies on predicting the toxicity of nanoparticles using ML models, including the results of the current article (green colored). R Concu et al. [37] developed a unified quantitative-structure activity/toxicity relationships (QSAR/QSTR) model based on artificial neural networks to predict general toxicity profiles (toxic cases and non-toxic cases) of 260 NPs. The Y-randomization approach was applied to check the robustness of the present ANN model. A K Halder et al. [38] built a perturbation theory machine learning-based QSTR model for predicting binary genotoxicity of metal oxide nanoparticles. External validation of the proposed model was performed for seven different metal oxides: Fe_3_O_4_, CuO, Fe_2_O_3_, ZnO, SiO_2_, TiO_2_, and Co_3_O_4_.

Of particular interest are studies of predicting the dynamic toxicity of nanoparticles using modern machine learning methods. K Song et al. [39] investigated the dynamic cytotoxicity of zinc (ZnO) nanoparticles for the bacterium *E. coli*. E He et al. [40] study the influence of Zn concentration in an organism to predict the dynamic toxicity of ZnCl_2_. JM Seiffert et al. [41] describe the use of an impedance spectroscopy approach to study the dynamic toxicity of metal oxide nanoparticles (ZnO, CuO, and TiO_2_). M Tarantola et al. [42] monitor cell dynamics as a measure of gold (Au) nanoparticle toxicity. AD Gholap et al. [43] review the application of artificial intelligence and machine learning tools in the biomedical field and note the enormous potential for accelerating development in the healthcare sector.

An analysis of state-of-the-art publications shows the relevance of using modern machine learning methods and artificial intelligence to study the toxicity of nanoparticles. To the best of our knowledge, there is currently virtually no research on the use of advanced ML models to predict the dynamic toxicity of nanoparticles based on their concentration in nutrient solution. Most relevant studies only consider the problem of binary classification of toxicity. Our main contributions in this article are as follows:−We performed a comparative analysis of the current state of research in the field of nanoparticle toxicity analysis;−We trained a regression model for predicting the toxicity of nanoparticles depending on their concentration in the nutrient solution at a fixed point in time of the stationary growth phase;−We trained a classification model for predicting the toxicity class of nanoparticles depending on their concentration in the nutrient solution at a fixed point in time of the stationary growth phase;−We carried out an analysis of the significance of physicochemical descriptors for models’ prediction results;−We have conducted extensive experiments with our own dataset, and the results show that the models proposed can be used in toxicity analyses of different kinds of nanoparticles for *E. coli* bacteria.

The rest of this paper is organized as follows: Section 2 provides a nanoparticle toxicity prediction problem statement. Section 3 presents dataset preparation and preliminary study of regression and classification models. Section 4 presents experimental results on data obtained. Section 5 summarizes the discussion.

## 2. Problem Statement

### 2.1. Toxicity Prediction Problem

To formally describe the mathematical formulation of the problem of predicting the toxicity of a nanoparticle *NP*, we introduce the following notation:

*X*—set of objects (nanoparticles NP);

*Y*—set of answers (cell viability index $NP_{via}$ or toxicity index $NP_{tox}$);

y:X→Y—unknown dependency (target function).

It is necessary to find an algorithm a:X→Y, a decision function that approximates *y* on the entire set *X*.

Cell viability index $NP_{via}$ describes the percentage of functioning and living cells, while toxicity index $NP_{tox}$ refers to the ability of a substance to damage and/or kill cells. The toxicity of a nanoparticle and cell viability are formally related to each other by an inverse relationship:(1)$$NP_{tox}=100-NP_{via}$$

Cell viability index $NP_{via}$ is calculated using the formula:(2)$$NP_{via}=\frac{I}{I_{0}}\cdot 100$$
where $I_{0}$ and *I*—intensity of bioluminescence of control (in a solution with zero concentration of nanoparticles) and experiment (in a solution containing nanoparticles) at a fixed exposure time of the test sample with the biotest.

#### 2.1.1. Regression Problem

In general, the problem of quantitative prediction of the toxicity of a nanoparticle is formulated as follows:

Let ${\left\{(x_{i},y_{i})\right\}}_{i=1}^{l}\subset X$ be the training sample, Y=R, where $x_{i}=(x_{i1},x_{i2},\ldots ,x_{in})$ is the feature vector for the *i*-th object, characterizing the physicochemical parameters of nanoparticles, as well as the concentration of nanoparticles in the solution, and the time from the start of the experiment, $y_{i}=y(x_{i}),i=1,\ldots ,l$ is the target value for the *i*-th object (viability index $NP_{via}$), *l*—number of objects in the training sample, *n*—number of features.

The loss function is defined as:(3)$$L(X,y,a)=\frac{1}{n}\sum_{i=1}^{l}{\left(y_{i}-a(x_{i})\right)}^{2}.$$

It is necessary to find an algorithm a(x), that minimizes the error between the predicted values ${\hat{y}}_{i}=a(x_{i})$ and the real values *y_i_* (minimizes the loss function):(4)$$\hat{a}=\arg\;\underset{a}{\min}\sum_{i=1}^{l}{\left(y_{i}-a(x_{i})\right)}^{2}.$$

To find such an algorithm, one can use models based on the implementation of gradient boosting on decision trees. The model builds an ensemble of many decision trees, where each subsequent tree is trained on the errors of previous trees. Training occurs in several stages, where at each stage a new tree $b_{t}(x),$ is added to minimize the error, $a_{t+1}=a_{t}+\eta \cdot b_{t}(x),$ where ${\hat{b}}_{t}$ is the current ensemble prediction, η is the learning rate, and $b_{t}(x)$ is the new tree trained on the current residuals.

The final model $\hat{a}(x)$ is the sum of all trees trained at various stages:(5)$$\hat{a}(x)=\sum_{t=1}^{T}\eta \cdot b_{t}(x),$$
where *T* is the total number of trees, and $b_{t}(x)$ is the prediction of the *t*-th tree.

Each tree $b_{t}(x)$ is a piecewise constant function defined by the vertices of the decision tree. The tree divides the feature space into many areas and assigns a certain prediction value to each area. Let us denote these areas as $R_{tj}$ for the *j*-th vertex of the tree *t*, and the prediction value for this area as $w_{tj}$. Then the prediction of the *t*-th tree for the feature vector *x* can be written as:(6)$$b_{t}(x)=\sum_{j=1}^{J_{t}}w_{tj}\cdot I[x\in R_{tj}],$$
where $J_{t}$ is the number of leaves in the *t*-th tree, $w_{tj}$ is the prediction value for the *j*-th area in the *t*-th tree, $I[x\in R_{tj}]$ is an indicator function that takes the value 1 if *x* falls into the area $R_{tj}$ and 0 otherwise.

#### 2.1.2. Classification Problem

The problem of predicting the toxicity class of a nanoparticle is formulated as follows: Let ${\left\{(x_{i},y_{i})\right\}}_{i=1}^{l}\subset X$ be the training sample, Y={1,…,M} be the set of labels of non-overlapping toxicity classes, $y_{i}=y(x_{i}),\;i=1,\ldots ,l$ be the target value for the *i*-th object (class label $\bar{1,M}$), *l* be the number of objects in the training set, *n* be the number of features.

It is necessary to find an algorithm a(x), that minimizes the loss function:(7)$$L(X,y,a)=-\frac{1}{l}\sum_{i=1}^{l}\sum_{j=1}^{M}y_{ij}\log a_{ij}(x).$$

### 2.2. Evaluation Metrics

The following metrics are commonly used to evaluate ML models. Metrics for evaluating regression models are:(1)Mean Absolute Error, MAE
(8)$$MAE=\frac{1}{l}\sum_{i=1}^{l}\left|y_{i}-{\hat{y}}_{i}\right|.$$(2)Mean Squared Error, MSE
(9)$$MSE=\frac{1}{l}\sum_{i=1}^{l}(y_{i}-{\hat{y}}_{i}{)}^{2}.$$(3)Root Mean Squared Error, RMSE
(10)$$RMSE=\sqrt{\frac{1}{l}\sum_{i=1}^{l}(y_{i}-{\hat{y}}_{i}{)}^{2}}.$$(4)Coefficient of determination, R^2^
(11)$$R^{2}=1-\frac{\sum_{i=1}^{l}(y_{i}-{\hat{y}}_{i}{)}^{2}}{\sum_{i=1}^{l}(y_{i}-{\bar{y}}_{i}{)}^{2}}.$$(5)Predictive relevance, Q^2^
(12)$$Q^{2}=1-\frac{\sum_{i=1}^{l}(y_{i}-{\tilde{y}}_{i}{)}^{2}}{\sum_{i=1}^{l}(y_{i}-{\bar{y}}_{i}{)}^{2}}.$$


In (3)–(7) $y_{i}$ defines real values, ${\hat{y}}_{i}$ defines predicted values on train data, ${\tilde{y}}_{i}$ defines predicted values on test data, ${\bar{y}}_{i}$ is the mean of the real values, ${\bar{y}}_{i}=\sum_{i=1}^{l}y_{i}$.

Metrics for evaluating classification models are:
(1)Accuracy
(13)$$accuracy=\frac{TP+TN}{TP+TN+FP+FN}$$(2)Precision
(14)$$precision=\frac{TP}{TP+FP}$$(3)Recall (Sensitivity)
(15)$$recall=\frac{TP}{TP+FN}$$(4)F1-score (F-measure)
(16)$$F1=\frac{2\times precision\times recall}{precision+recall}$$(5)Specificity
(17)$$specificity=\frac{TN}{TN+FP},$$
where TP (True Positive)—the classifier correctly states that the object belongs to the class under consideration, TN (True Negative)—the classifier correctly states that the object does not belong to the class under consideration, FP (False Positive)—the classifier incorrectly states that the object belongs to the class under consideration, FN (False Negative)—the classifier incorrectly states that the object does not belong to the class under consideration.


To thoroughly evaluate the multiclass classification model, let us consider one more additional metric:
(6)Log Loss
(18)$$Logloss=-\frac{1}{l}\sum_{i=1}^{l}\sum_{j=1}^{M}y_{ij}\ln a_{ij},$$
where *l* is the number of observations, *M* is the number of classes, $a_{ij}$ is the answer (probability) of the algorithm on the *i*-th object to the question of whether it belongs to the *j*-th class, $y_{ij}=1$, if *i*-th object belongs to the *j*-th class, $y_{ij}=0$ otherwise.


## 3. Materials and Methods

### 3.1. Nanoparticle Representation

Equipment: Tecan Infinite F200 Fluorescence Microplate Reader.

Laboratory: Federal Research Centre of Biological Systems and Agrotechnologies of the Russian Academy of Sciences, Center for Nanotechnologies in Agriculture

Analysis: Determination of biotoxicity in the bacterial luminescence inhibition test.

The biotoxicity of nanoparticles was assessed using a constitutively luminescent strain of Escherichia coli K12 TG1 (commercial name “Ecolum”, “NVO IMMUNOTECH”, Russia), carrying a hybrid plasmid pUC19 with cloned luxCDABE genes of Photobacterium leiognathi 54D10. To do this, 5 mL of distilled water cooled to 4 ˚C and 5 mL of room temperature distilled water were added to the initial lyophilisate, shaken vigorously until the precipitate was completely dissolved, and kept in the refrigerator for 30 min. After this, 100 μL of the resulting suspension was added to a 96-well culture plate to previously prepared dilutions of ZnO (6000 nm; 87.8 mV), Fe_3_O_4_ (1528.3 nm; 9.2 mV), and SiO_2_ (262.2 nm; 60.9 mV) nanoparticles obtained chemically in the Laboratory of Nanostructure Synthesis of Orenburg State University. Thus, the final series of concentrations ranged from 100 to 1.2 × 10^−5^ mg/mL. The plate with experimental samples was placed in the device, recording the luminescence intensity in relative luminescence units (RLU) every 5 min during 3 h of exposure. At the same time, effective inhibitory concentrations 80 (EC80) were determined; 50 (EC_50_) and 20 (EC_20_)% glow, i.e., toxic, conditionally toxic, and sub-inhibitory doses, respectively. For each type of nanoparticle, an assessment of the suspension was performed, which is an integral indicator at a given concentration. The size and zeta potential of nanoparticles were calculated based on three measurements as an average.

### 3.2. Dataset Preparation

As a result of a series of experiments, the data from the laboratory were provided in *.xls format, in the form of a two-dimensional table, in which the rows corresponded to the observation time with an interval of 5 min, and the columns corresponded to the concentration of nanoparticles in the nutrient solution; the table cells presented data on bioluminescence taken from the device. In this study, the use of periodic table-based descriptors is not suitable since the properties of nanoparticles strongly depend on their size and are largely associated with the state of the boundary phase of the solid particle—dispersed medium. Therefore, integral descriptors are more suitable in this case. For periodic table-based descriptors, it is also necessary to proceed from the solubility of the nanoparticle, which is difficult due to the lack of an equipment base. For further use of the provided data and training of ML models, the experimental data is converted into a flat table. To describe the physical properties of the nanomaterial, additional information about the sizes of nanoparticles and their zeta potential was used; the data for each nanoparticle amounted to 888 records. The size of the final LumenTox dataset is 2664 records. The obtained physicochemical features are presented in Table 2. A fragment of the LumenTox dataset for training ML models is presented in Figure 1. The traditional machine learning models are chosen over deep learning models due to the relatively small size of the dataset and the possibility of overfitting.

### 3.3. Dynamic Nanotoxicity Regression Problem

We compared four regression models on the LumenTox dataset. The results for metrics (10) and (11) are presented in Table 3. All four models show relatively good prediction results.

Let us choose LightGBMRegressor as the main regression model. For such a model, one can visualize ensemble decision trees using the plot_tree (https://lightgbm.readthedocs.io/en/latest/pythonapi/lightgbm.plot_tree.html) (accessed on 26 September 2024) method. The train/test split ratio used for this study is 80-20. The 80-20 ratio allows for sufficient data to train the model while still allowing for the ability to test its performance on an independent dataset. The selection of hyperparameters is carried out using the software platform for automatic optimization of hyperparameters Optuna [44]. We obtain the following optimal regressor parameters:

{‘n_estimators’: 1200, ‘bagging_freq’: 11, ‘learning_rate’: 0.0098, ‘num_leaves’: 597, ‘subsample’: 0.5782, ‘colsample_bytree’: 0.0838, ‘min_data_in_leaf’: 4, ‘max_depth’: 56, ‘lambda_l1’: 2, ‘lambda_l2’: 6, ‘min_gain_to_split’: 1.5627, ‘bagging_fraction’: 0.7426, ‘feature_fraction’: 0.8956}

Having found optimal hyperparameters, the metric value RMSE = 2.26 decreased to RMSE = 1.48. To track the quantitative indicator of the dynamics of the toxicity of a nanoparticle, we will fix the time of 90 min and vary the concentration in the range from 100 g/L to 0.000012 g/L on a uniform logarithmic scale; each subsequent concentration value is 2 times less than the previous one. The time of 90 min is the time of the stationary growth phase when maximum luminescence is observed in the control on the plateau. Experimental results on metrics (9) and (10) are presented in Section 4.1 of the article.

### 3.4. Nanotoxicity Classification Problem

Let us compare four classification models on the LumenTox dataset. The results for metrics (18) and AUC score are presented in Table 4. All four models show relatively good prediction results.

Let us choose LightGBMClassifier as the main classification model. For such a model, one can visualize ensemble decision trees using the plot_tree method. The train/test split ratio used for this study is 80-20. To solve the problem of multiclass classification, we can identify seven main toxicity categories in descending order. Any threshold of concentrations from EC10 to EC90 can be identified, but usually these values are given since they correspond to toxic, semi-toxic, and sub-inhibitory doses. A description of each class and toxicity levels is presented in Table 5. Color differentiation of classes is done for better visual clarity. Let us add a new class column to the LumenTox dataset, containing the toxicity category for each initial toxicity value. Let us remove the features of electronegativity and ionic radius from the dataset since they are not significant by feature importance analysis (https://lightgbm.readthedocs.io/en/latest/pythonapi/lightgbm.plot_importance.html) (accessed on 26 September 2024).

The selection of hyperparameters is carried out using the software platform for automatic optimization of hyperparameters, Optuna. We obtain the following optimal classifier parameters:

{‘boosting_type’: ‘goss’, ‘num_iterations’: 450, ‘num_leaves’: 8, ‘max_depth’: 4, ‘learning_rate’: 0.1756, ‘scale_pos_weight’: 7, ‘colsample_bytree’: 0.8, ‘reg_alpha’: 3, ‘reg_lambda’: 8}

We use 10-fold cross-validation. To track the dynamics of the toxicity concentration class, we will fix the time of 90 min and vary the concentration in the range from 100 g/L to 0.000012 g/L on a uniform logarithmic scale; each subsequent concentration value is 2 times less than the previous one. Experimental results on metrics (13), (15), (16), and (18) are presented in Section 4.2 of the article.

## 4. Experimental Results

### 4.1. Results for Dynamic Nanotoxicity Regression Problem

Figure 2 presents the results of studies on the significance of physicochemical descriptors for predicting a regression model. Figure 2a shows the results of the Shapley additive explanatory model (SHAP), which calculates the contribution of each feature to the model’s output. Features with positive SHAP values have a positive effect on the forecast, while features with negative SHAP values have a negative impact on the forecast. The diameter of the nanoparticle, the concentration of nanoparticles in the nutrient solution, and time are the most significant features. Figure 2b shows a radar chart based on the significance of features for the LumenTox dataset and for data on each nanoparticle separately. For ZnO and Fe_3_O_4_ nanoparticles, an almost identical surface of the significance of the features is obtained, while for the SiO_2_ nanoparticle, the features of concentration and time are less significant for the forecast. Molecular weight, electronegativity, and ionic radius have virtually no effect on the predictive results of the regression model. For the LumenTox dataset, nanoparticle diameter and zeta potential have little effect on the prediction result; the features of nanoparticle concentration and time are the most significant.

Figure 3a–d shows scatterplots with a regression line for the LumenTox dataset and data for each nanoparticle; Figure 3e shows a residual plot without cross-validation; and Figure 3d shows a residual plot with cross-validation. From Figure 3a–d, we can conclude that the data has a strong positive linear relationship with no existence of outliers. From Figure 3e,f we can conclude that there are not any clear patterns in data distribution; points are scattered randomly around the residual line, which indicates good prediction results of the model.

Figure 4 shows a scatter plot of the test objects, which on the Y axis show the difference between the two maximum probabilities of the object class; the closer the values are to 1, the better.

Figure 5 shows a visualization (a) of the Cook’s Distance, (b) of the DFITS, and (c) of the Leverage. Cook’s Distance and DFITS Conclusions: Most of the points are concentrated around the Cook’s Distance/DFITS = 0 line, meaning that most of the observations do not have a significant impact on the model. The points have small Cook’s Distance/DFITS values, indicating that removing these observations will not change the predicted values of the model much. Leverage conclusions:Low model impact: Observations with low Leverage values have minimal impact on the model parameters, i.e., the model is mostly determined by the majority of the data rather than by individual extreme values.Model stability: Low Leverage values indicate that the model is likely stable and is not significantly affected by removing individual observations.No extreme values: Low Leverage values also indicate that there are no extreme values or outliers in the data that could significantly affect the regression results.Well distributed data: Such a Leverage distribution may indicate that the data is well distributed and does not contain significant anomalies.

Figure 6 shows the predicted results of the trained regression model compared to real laboratory studies of nanoparticle toxicity. It can be concluded that the predicted results of the trained regression model are very close to the real ones. An increase in the concentration of metallic nanoparticles leads to a decrease in the viability of microorganisms, which is due to the gradual degradation of nano- and microstructures in solution with the release of Zn^2+^ and Fe^2+^/Fe^3+^ ions [45]. The latter actively interact with various cellular structures, in particular with protein macromolecules and nucleic acids, which at high concentrations lead to disorganization of membrane complexes, disturbances of enzymatic and mitochondrial functions, increased synthesis of reactive oxygen species, oxidative stress, and ultimately cell death [46,47]. In this case, the chemical nature of the element in the nanoparticle and, accordingly, its reducing properties (chemical activity) are of particular importance. Thus, in the presented experiment, the dynamics of the luminescence of a bacterial strain when Fe_3_O_4_ nanoparticles were added to the nutrient solution were negative relative to the control until the concentration was reduced to 0.1 mg/mL; however, even after that, within a certain range, it remained at EC_20_ (a conditional plateau of the subinhibitory effect). ZnO nanoparticles, in turn, were toxic up to a threshold value of 9.5 × 10^−5^ mg/mL, and a sharp decline in viability occurred in the delta 10^−2^ ± 5 × 10^−2^ mg/mL, which indicates the impossibility of the adaptive mechanisms of prokaryotes to reduce metabolic load in this range. Thus, the Zn-containing nanoparticles showed significantly greater toxicity to Escherichia coli K12 TG1 than the magnetite-based nanoparticle. The optimal dosages recommended for further testing on animals as an alternative to antibiotic growth stimulants and a highly effective source of macro- and microelements were 9.5 × 10^−5^ mg/mL in the first case and 0.1 mg/mL in the second.

SiO_2_ nanoparticles, on the contrary, with increasing concentration intensified the luminescence of microorganisms, which reflects an increase in metabolic capabilities and cell viability. The latter is associated with the potential role of these non-metallic nanoparticles as stress protectors [48]. In this case, the intensifying effect may be due to the restructuring of the cell wall and membrane-mediated physiological cycles, including the absorption of nutrients, or an increase in quorum-dependent interactions. Silicon oxide, unlike metals, stimulates the glow of the strain (metabolic functions); therefore, the values do not decrease from 100 at a non-toxic level to 0 at a toxic level (metals), but on the contrary, they grow from 100% in the control and above to 200% in the experiment. This is probably due to the nature of the element and the possibility of its incorporation into cellular structures in molecular, not ionic, form. The optimal dosage is 1 mg/mL.

The final achieved metrics (9), (10) and (12) of the LightGBMRegressor model on the LumenTox dataset for the problem of predicting toxicity depending on the nanoparticle concentration in the nutrient solution are MSE = 2.19, RMSE = 1.48, and Q^2^ = 0.99. We can conclude the good predictive ability of the trained model.

### 4.2. Results for the Nanotoxicity Multi-Class Classification Problem

Figure 7 shows the normalized confusion matrix of the trained multi-class classifier (see Table 3). Toxicity classes Tox, NTOX, NTOX+, and NTOX2+ have the best forecast results, while EC20, EC80, and EC50 classes show slightly worse results.

Figure 8 shows the ROC curves and Precision-Recall curves for each toxicity class trained by the multiclass classifier. The ROC curve for each class is very close to the upper left corner, indicating the high predictive ability of the model. The average precision score for each class is close to one, which also indicates the high predictive ability of the model.

Figure 9 shows one of the decision trees of the trained multiclass classifier. To predict the toxicity class, the results of all decision trees are summarized using the loss function. Visualization of decision trees contributes to explainable AI and can be very useful for researchers in the field of biology.

The final achieved metrics (13), (15), (16), and (18) of the LightGBMClassifier model of the LumenTox dataset for the problem of predicting the toxicity class depending on the nanoparticle concentration in the nutrient solution are Accuracy = 0.9756, Recall = 0.9623, F1-Score = 0.9640, and LogLoss = 0.1855. We can conclude the good predictive ability of the trained model.

### 4.3. Applying Trained ML Models to Other Nanoparticles

We performed a study on the transferability of the trained classification model results to new data. An analysis of the toxicity class for Co_3_O_4_ and Mn_2_O_3_ nanoparticles was performed for a fixed time of 90 min and concentration in the range from 20 g/L to 0.0000024 g/L, studied in the laboratory. Tables with the results of predicting the toxicity class with predicted and real values for the three original nanoparticles SiO_2_, ZnO, and Fe_3_O_4_ and two additional nanoparticles Co_3_O_4_ and Mn_2_O_3_ are presented in the Appendix A. The coloring of toxicity classes was carried out in accordance with Table 5. Figure 10 shows the normalized confusion matrix of the trained multi-class classifier on Mn_2_O_3_ nanoparticles. Figure 11 shows the normalized confusion matrix of the trained multi-class classifier on Co_3_O_4_ nanoparticles. The initial laboratory data for Mn_2_O_3_ nanoparticles do not contain records corresponding to the Tox and NTOX2+ classes. The model shows fairly good prediction results for the EC80 class. The initial laboratory data for Co_3_O_4_ nanoparticles do not contain records corresponding to the NTOX2+ class. The model shows fairly good prediction results for the NTOX class.

For the Mn_2_O_3_ nanoparticle, a relatively good prediction was obtained for the optimal toxicity dosage; for the Co_3_O_4_ nanoparticle, a relatively good prediction was obtained for low concentrations of the nanoparticles and a poor prediction for large concentrations of the nanoparticles. It is recommended to further train and fit the ML models using new data from additional nanoparticles.

## 5. Discussion

The article provides a comparative analysis of the current state of research in the field of analyzing the toxicity of nanoparticles and the use of modern machine learning methods to solve the problem.

One of the main challenges in studying the toxicity of nanoparticles is the large variety of nanoparticles and, at the same time, the lack of data for training ML models. Thanks to the laboratory studies of SiO_2_, ZnO, and Fe_3_O_4_ nanoparticles assessed using the constitutively luminescent *Escherichia coli K12* strain, we created two new datasets for studying the dynamic toxicity of nanoparticles based on physicochemical attributes for regression and classification problems, which provides added value to the existing pool of nanoparticle data.

A regression model was trained to predict the quantitative toxicity of a nanoparticle depending on the nanoparticle’s concentration in the nutrient solution at a fixed point in time. A multiclass classification model was trained to predict the toxicity class of a nanoparticle depending on the nanoparticle’s concentration in the nutrient solution at a fixed point in time. An analysis of the significance of physicochemical descriptors for models’ prediction results was carried out, and it was concluded that the diameter of the nanoparticle and the concentration in the nutrient solution are the most significant features of predictive models. Experiments were carried out on the use of the classification model on two additional nanoparticles, Co_3_O_4_ and Mn_2_O_3_; the experiments show relatively good predictive results on training data and for additional nanoparticles not used in training ML models.

Statistical analysis conducted, visualization of trained ML models, and analysis of metrics on validation data allow us to conclude that the models have a high predictive ability with no extreme values or outliers in the data. The data obtained is well distributed and does not contain significant anomalies. The developed ML models show good predictive results for nanoparticles that belong to the group of transition metals of the fourth period, which allows the trained models to be used to predict the toxicity of microelements from this group after additional training of the models on new data.

The regression model’s metrics achieved in other studies [27,30,33,35] are consistent with results obtained in this study with MSE = 2.19, RMSE = 1.48, and Q^2^ = 0.99. The classification model’s metrics achieved in other studies [28,29,32] are consistent with results obtained in this study with Accuracy = 0.97, Recall = 0.96, F1-Score = 0.96, and LogLoss = 0.18. The trained multi-class classifier is more flexible to determine the toxicity levels across seven classes compared to binary classification. One of the findings is that ZnO nanoparticles are highly toxic to *Escherichia coli*, which is consistent with the conclusion drawn in the study [29].

The use of modern machine learning models can significantly reduce the number of animals introduced into experiments when testing new drugs and reduce time and labor costs, thereby helping to improve the production process. Moreover, computer models can be used to assess environmental pollution by technogenic nanostructures and predict possible negative effects. In the presented experiment, the optimal dosages for the studied nanoparticles were as follows: ZnO nanoparticle—9.5 × 10^−5^ mg/mL; Fe_3_O_4_ nanoparticle—0.1 mg/mL; SiO_2_ nanoparticle—1 mg/mL. The latter can be used in animal husbandry in the case of metal nanoparticles as an alternative to antibiotics or a source of essential elements, or in crop production—in the case of SiO_2_ nanoparticles—as stress protectors.

The obtained results of studies on the analysis of toxicity of nanoparticles on bacteria can be used by scientists conducting similar studies on prokaryotes [29,30,34]. Special attention can be paid to the study of the dynamics of toxicity depending on the concentration of the substance in the nutrient solution, which is a significant difference of this study. Implementation of proposed toxicity prediction models in cellular and animal studies is planned for next year.

## 6. Limitations

The research presented in this article has the following limitations: The ML models were trained on data obtained from the laboratory analysis of three nanoparticles SiO_2_, ZnO, Fe_3_O_4_ on the *Escherichia coli K12 TG1* bacterium using the method of bacterial luminescence inhibition. LightGBMRegressor and LightGBMClassifier models are selected as regression and classification models, respectively. The selection of model hyperparameters was carried out using the Optuna library. The following physicochemical descriptors were used to predict toxicity: concentration, diameter, zeta potential, molecular weight, electronegativity, and ionic radius. The classification model was further tested on data obtained from two additional nanoparticles, Co_3_O_4_ and Mn_2_O_3_. Potential improvements to trained models come from adding new training data obtained from new types of nanoparticles. Good predictive results were obtained for nanoparticles that belong to the group of transition metals of the fourth period. We see great potential for the applicability of the employed approach to predict the toxicity of microelements from other groups after additional training on new data. This study is focused on building a database of toxic effects on prokaryotes. Experiments on eukaryotic cells and animals are planned for next year to test subinhibitory doses.

## Figures and Tables

**Figure 1 toxics-12-00750-f001:**
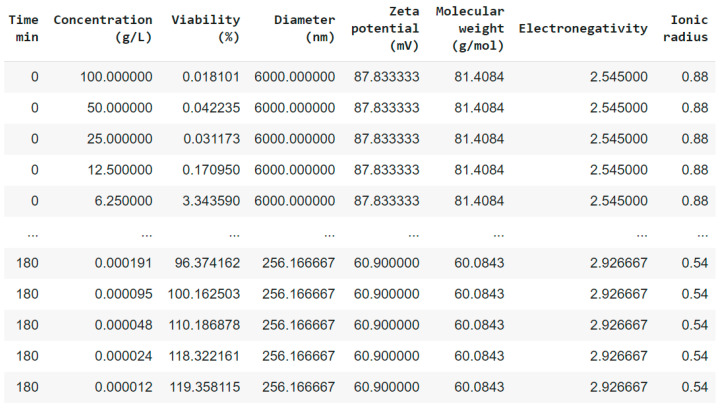
Fragment of the obtained LumenTox dataset.

**Figure 2 toxics-12-00750-f002:**
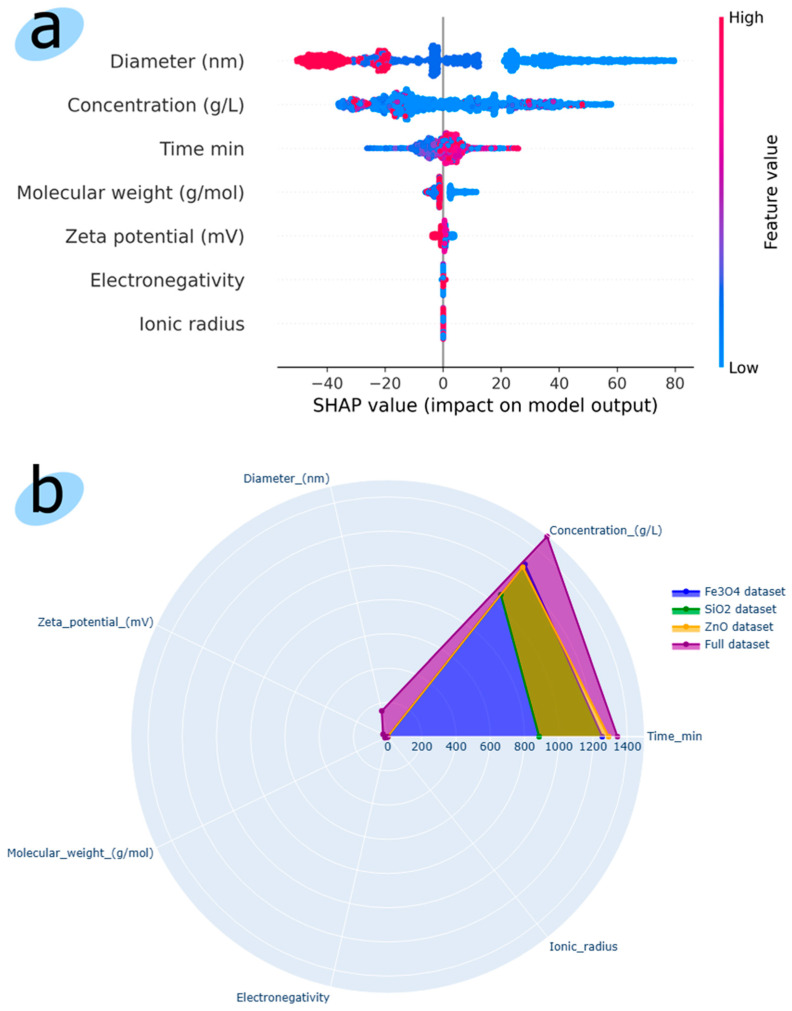
(**a**) SHAP model of each feature contribution to the model outcome (**b**) radar chart comparing seven physicochemical attributes of three nanoparticles and the full LumenTox dataset.

**Figure 3 toxics-12-00750-f003:**
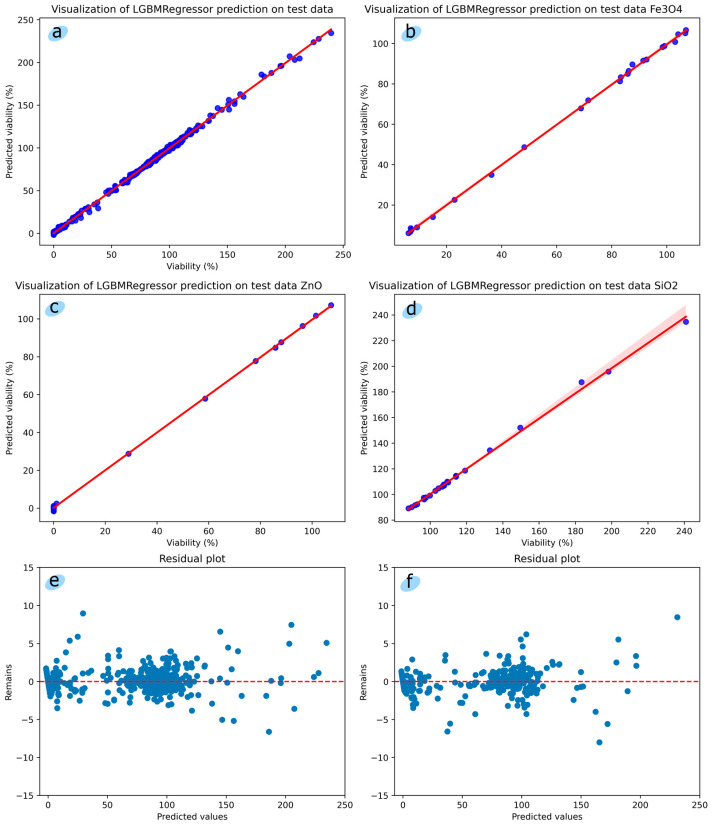
(**a**–**d**) scatter plots with regression line, (**e**) residual plot without cross-validation, (**f**) residual plot with cross-validation. The horizontal line *y* = 0 is a line representing zero error. The blue dots represent data points, each data point has one residual, i.e. vertical distance between a data point and the regression line.

**Figure 4 toxics-12-00750-f004:**
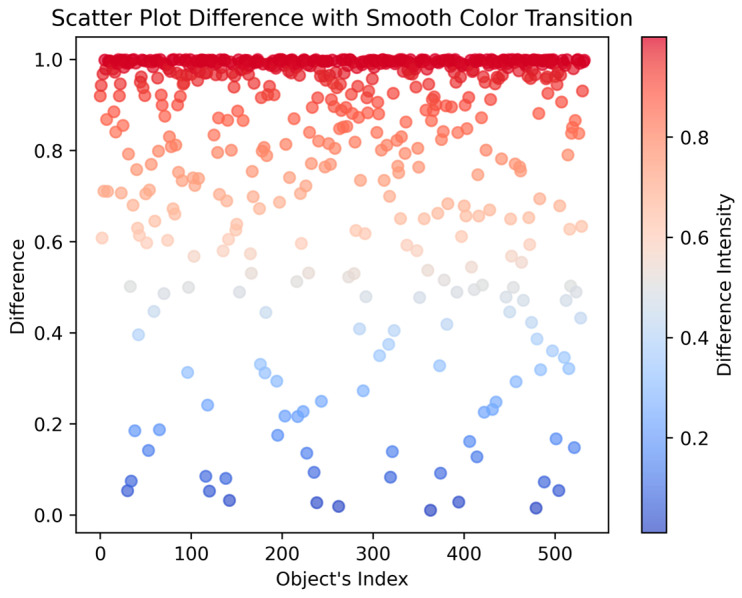
Scatter plot difference between the two maximum probabilities of the object class.

**Figure 5 toxics-12-00750-f005:**
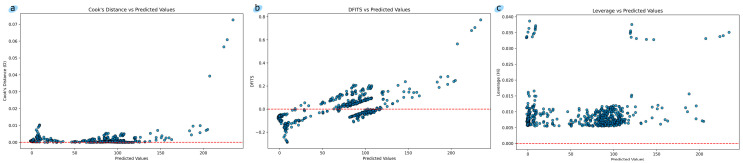
(**a**) Cook’s Distance vs. predicted values; (**b**) DFITS vs. predicted values; and (**c**) Leverage vs. predicted values. For the Cook’s Distance the horizontal line is a line representing threshold below which objects are considered to have little influence on the model. Each blue dot displays the Cook’s Distance value for a specific observation. For the DFITS the horizontal line is a line representing threshold, the point where DFITS values close to 0 indicate that the observation has little influence on the model. Each blue dot displays the DFITS value for a specific observation. For the Leverage the horizontal line is a line representing the baseline, Leverage determines how “far” the observation is from the rest of the predictors. Each blue dot displays the Leverage value for a specific observation.

**Figure 6 toxics-12-00750-f006:**
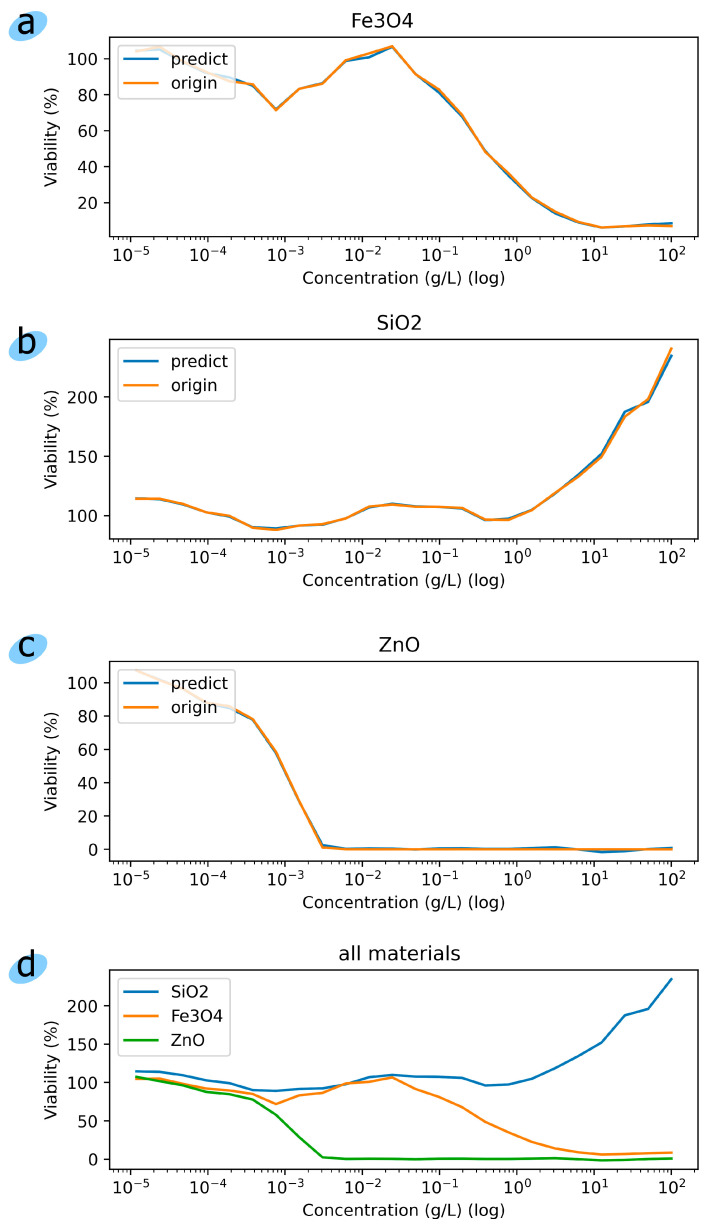
(**a**–**c**) graphs of cell viability depending on the concentration of nanoparticles in the nutrient solution; (**d**) dynamics of toxicity for nanoparticles SiO_2_, Fe_3_O_4_, and и ZnO.

**Figure 7 toxics-12-00750-f007:**
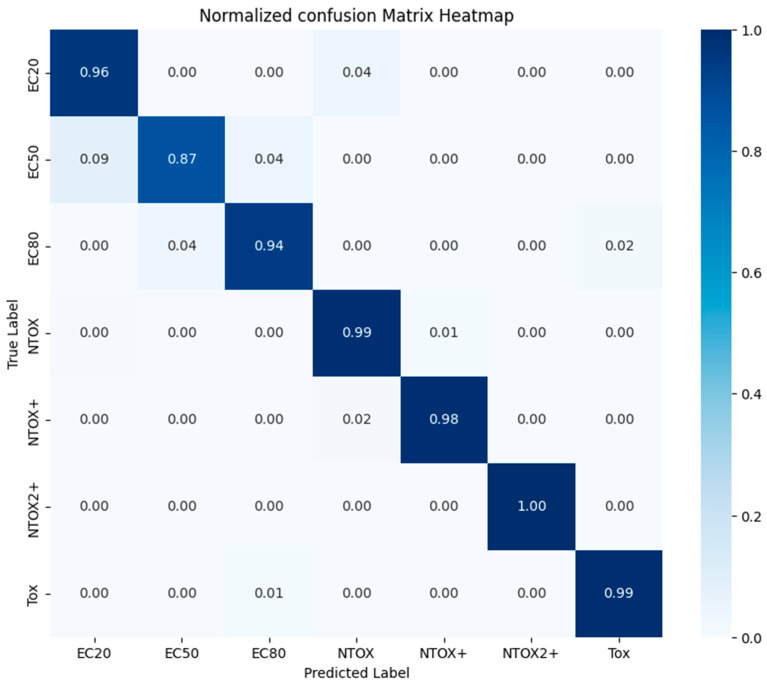
Normalized confusion matrix of trained LightGBMClassifier to predict toxicity class.

**Figure 8 toxics-12-00750-f008:**
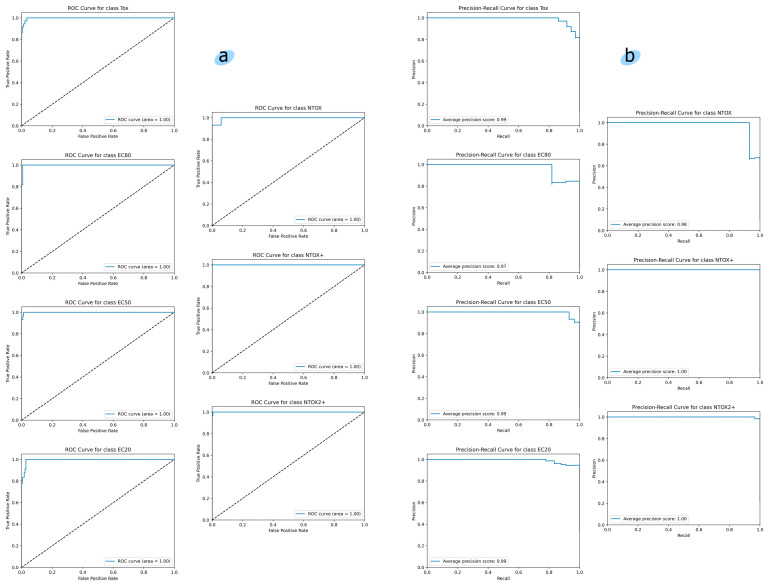
(**a**) ROC curves and (**b**) Precision-Recall curves of seven classes of toxicity. Dashed line represents the ROC curve for a random guess.

**Figure 9 toxics-12-00750-f009:**
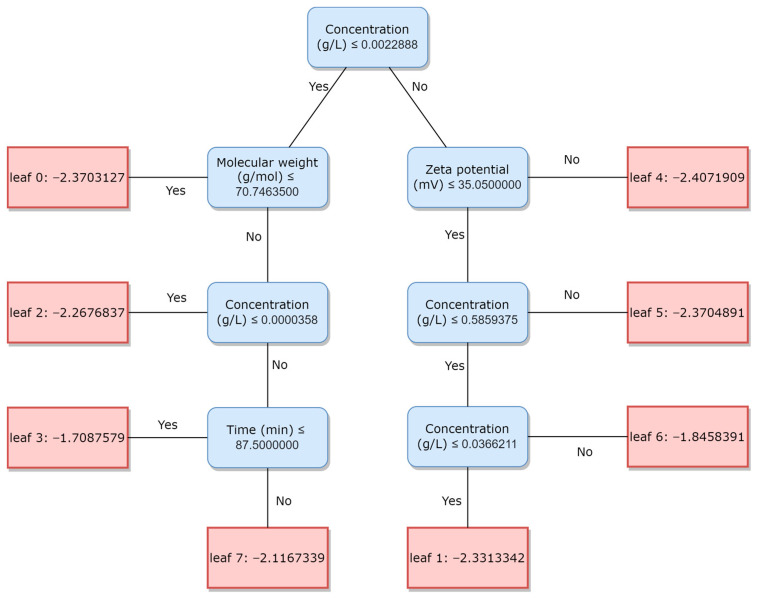
Visualization of one of the decision trees for the trained LGBMClassifier model.

**Figure 10 toxics-12-00750-f010:**
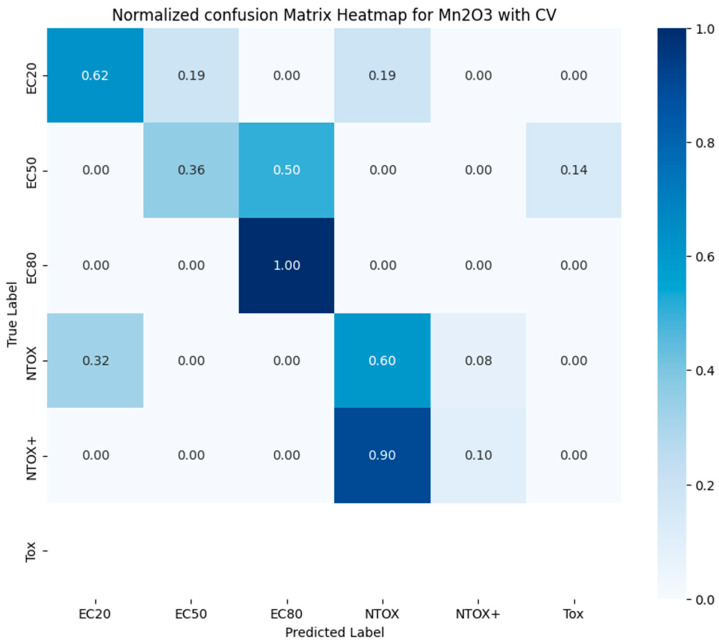
Normalized confusion matrix of trained LightGBMClassifier to predict toxicity class of new Mn_2_O_3_ nanoparticle.

**Figure 11 toxics-12-00750-f011:**
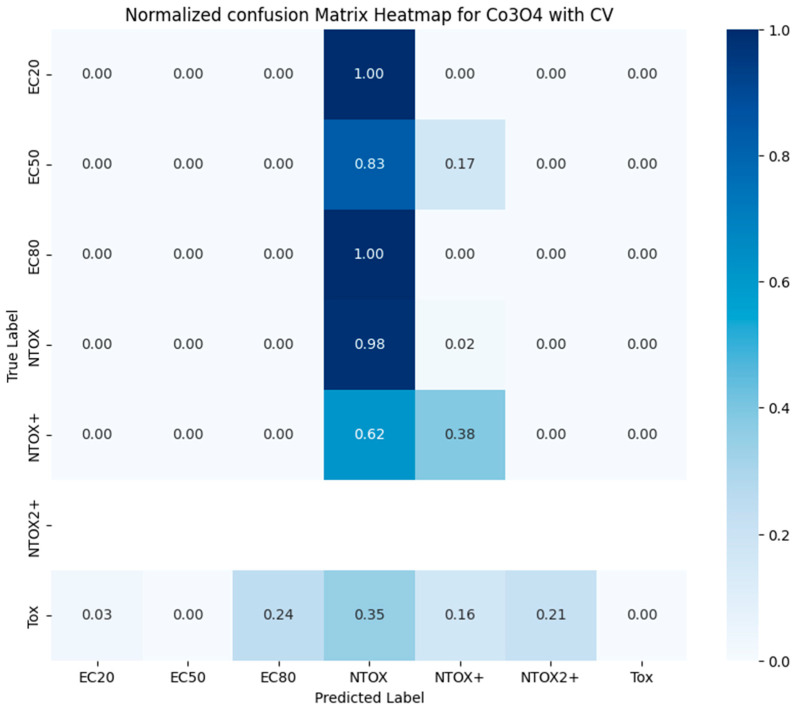
Normalized confusion matrix of trained LightGBMClassifier to predict toxicity class of new Co_3_O_4_ nanoparticle.

**Table 1 toxics-12-00750-t001:** Studies on nanoparticle toxicity prediction evaluated in different nutrient solutions.

Type(s) of NPs	Physicochemical Attributes/Descriptors	Environment (Nutrient Solution)	Data Size (Rows)	Main Outcomes	Ref.
Ag, Al, Au, C, Cd, Ce, Co, Cu, Fe, Mg, Mn, Ni, Pt, Se, Si, Ti, Zn, and Zr	Diameter, zeta potential, cell type, concentration, coat, line primary cell, cell morphology, cell age, cell organ, molecular weight, electronegativity, and ionic radius	Catfish, hamster, human, monkey, mouse, pig, rabbit, and Rat	3088	Quantitative prediction of cytotoxicity in vitro: 10-foldCV Q^2^ = 0.86 RMSE = 12.2%	[27]
ZnO, TiO_2_, SiO_2_, Fe_3_O_4_, Al_2_O_3_, CuO, and Fe_2_O_3_	Core size, hydrodynamic size, surface charge, specific surface area, formation enthalpy, conduction band energy, valence band energy, electronegativity, exposure time, and dose	Human, hamster, and mouse	575	Voting ensemble binary classification of cytotoxicity in vitro: Accuracy = 92%	[28]
Er_2_O_3_, Gd_2_O_3_, CeO_2_, Co_2_O_3_, Mn_2_O_3_, Co_3_O_4_, and Fe_3_O_4_/WO_3_	Molecular weight, electronegativity, oxidation number, atomic number, valence electron, period number, core environment, zeta-potential, surface area, hydrodynamic size, and particle size	*E. coli*	25	Binary classification of cytotoxicity in silico: Precision = 75% Accuracy = 87.5% F-measure = 85.7%	[29]
CuO, ZnO, TiO_2_, ZrO_2_	Primary size, surface area, zeta potential, and hydrodynamic diameter	*E. coli*	22	Quantitative prediction of toxicity: R^2^ = 0.911, RMSE = 0.091 MAE = 0.067	[30]
Ag_2_O, Al, Au, Bi_2_O_3_, CaO, Co_2_O_3_, Er_2_O_3_, Fe_3_O_4_, Ga_2_O_3_, HfO_2_, MoO_3_, Ni_2_O_3_, Pd, Sn, V2O_3_, W, Yb_2_O_3_,Zn	Hydration enthalpy, sum of electrons of the metals and semimetals, electrons of the metals, and metal ionization potential	Human, mammal, bacteria, crustaceans, fish, plants, and eukaryotes	935	Binary classification of toxicity: Accuracy = 91.7% Precision = 93.5% Sensitivity = 95.3% Selectivity = 86%	[32]
TiO_2_	Additive electronegativity, electron affinity, surface area, and formation energy	Hamster	29	Quantitative prediction of cytotoxicity in vitro: R^2^ = 0.97	[33]
SiO_2_, TiO_2_, ZnO	Molecular weight, diameter, purity, surface charge, zeta potential, and hydrodynamic size	*Aliivibrio fischeri*	93	Quantitative prediction of toxicity: FIT = 4.409	[34]
Ag	Reducing agent, cell lines, exposure time, particle size, hydrodynamic diameter, zeta potential, wavelength, and concentration	Rat	1315	Quantitative prediction of toxicity: MSE = 17.83 RMSE = 4.22 MAE = 2.49 R^2^ = 0.97	[35]
FeO, SiO_2_, TiO_2_, Ag, CuO, ZnO, GO, MnO, SWCNT	Shape, zeta potential, hydro_size, primary size, and surface area	Human, rat, and mouse	1498	Binary classification of toxicity: Accuracy = 72%	[36]
SiO_2_, ZnO, Fe_3_O_4_	Concentration, diameter, zeta potential, molecular weight, electronegativity, and ionic radius	*E. coli*	2664	Quantitative prediction of dynamic toxicity: 8-foldCV MSE = 2.19 RMSE = 1.48 Q^2^ = 0.99 Multi-class classification of dynamic toxicity: 10-foldCV Log Loss = 0.1855 Accuracy = 0.9756 Recall = 0.9623 F1-Score = 0.9640	Our research

**Table 2 toxics-12-00750-t002:** Physicochemical attributes of the LumenTox dataset.

Attribute	Description	Unit of Measurement
Concentration	Concentration of nanoparticles in nutrient solution	mg/mL
Diameter	Diameter of nanoparticles	nm
Zeta potential	Zeta potential of nanoparticles	mV
Molecular weight	Molecular weight of nanoparticles	g/mol
Electronegativity	Electronegativity of nanoparticle	tabular value
Ionic radius	Ionic radius of nanoparticles	tabular value
Time	Object observation time	min
Viability	Viability of nanoparticles	%

**Table 3 toxics-12-00750-t003:** Comparison of regression models for predicting nanoparticle toxicity on the LumenTox dataset.

Regression Model	R^2^	RMSE
ExtraTreesRegressor	0.9994	1.3193
LGBMRegressor	0.9977	2.2598
RandomForestRegressor	0.9990	2.2031
XGBRegressor	0.9992	1.8475

**Table 4 toxics-12-00750-t004:** Comparison of classification models for predicting nanoparticle toxicity on the LumenTox dataset.

Classification Model	Log Loss	AUC Score
ExtraTreesClassifier	0.2324	0.9935
LGBMClassifier	0.1011	0.9991
RandomForestClassifier	0.1289	0.9990
XGBClassifier	0.0909	0.9990

**Table 5 toxics-12-00750-t005:** Color-coded classes of nanoparticle toxicity concentration in nutrient solution.

Notation	Description	Toxicity Range	Number of Input Samples
Tox	Lethal concentration	95≤toxicity<100	556
EC80	Lethal concentration	80≤toxicity<95	249
EC50	Semi-lethal concentration	50≤toxicity<80	155
EC20	Sub-inhibitory concentration	20≤toxicity<50	278
NTOX	Non-toxic concentration	−5≤toxicity<20	998
NTOX+	Stimulating concentration	−30≤toxicity<−5	276
NTOX2+	Stimulating concentration	toxicity<−30	152

## Data Availability

The full dataset presented in this article is not readily available because the data are part of an ongoing study. Trained pkl models and sample datasets are publicly available: https://drive.google.com/drive/folders/1K-UQEXoLxM-chgw1GJt6NsuBvxDrm5UU. (accessed on 26 September 2024).

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
