# Peer review of "Prediction of Dynamic Toxicity of Nanoparticles Using Machine Learning"

_toxics, 2024, doi:10.3390/toxics12100750_

Round 1
Reviewer 1 Report
Comments and Suggestions for Authors
Please see file enclosed.

Author Response
Comments 1: Introduction – The authors have extensively covered earlier ML work aimed at targeting the prediction for the toxicity of nanoparticles. However, the authors disregard earlier multitarget classification work based on the Box-Jenkins moving average approach, either along with perturbation theory or not, which allow mining the response data pertaining to diverse experimental and/or theoretical conditions and can even incorporate experimental confidence assays information (see e.g.: Refs. [1, 2]). Indeed, these studies have shown to be quite efficient for predicting nanoparticles' toxicity. Not acknowledging existing multitarget ML modelling studies raises concerns about the inclusiveness of the analysis and the potential for alternative approaches.
Response 1: Thank you for pointing this out. Suggested publications added to Introduction section, new text highlighted in green color.
Comments 2: Data set preparation – Molecular descriptors are the core features in determining the performance of any ML model. Therefore, the reasons for not embarking on employing more simple descriptors, such as simply periodic table-based descriptors (see e.g. Ref. [3]), should be discussed, especially taken into account the forward SHAP results. It is also not clear whether some of the descriptors used to describe the physicochemical properties of the nanoparticles (such as, the ionic radius and electronegativity) have been measured or not? Please clarify.
Response 2: In this study the use of periodic table-based descriptors will not be suitable, since the properties of nanoparticles strongly depend on their size and are largely associated with the state of the boundary phase of the solid particle - dispersed medium. Therefore, integral descriptors are more suitable in this case. For periodic table-based descriptors, it is also necessary to proceed from the solubility of the nanoparticle, which is difficult due to the lack of an equipment base. For the both ionic radius and electronegativity their values were taken from corresponding tables.
Comments 3: Page 8, line 21: In fact, neural networks, especially deep learning models, are known for their ability to learn complex patterns from large amounts of data. However, when presented with limited data, they can easily over fit, meaning they learn the training data too well and struggle to generalise to new, unseen data. That is, they are prone to overfitting not underfitting as it is stated (Please correct.).
Response 3: Thank you for pointing this out. Text is corrected.
Comments 4: Dynamic nanotoxicity regression problem – The reasons for choosing the particular train / test split should be better discussed and justified. In addition, the values for Q 2 should be given to access the degree of possible overfitting.
Response 4: The 80/20 ratio allows for sufficient data to train the model while still allowing for the ability to test its performance on an independent dataset. Q2 metric calculated and added to the text. Obtained metric value is Q2 = 0.99907.
Comments 5: Results for dynamic nanotoxicity regression problem – While residual plots are a valuable tool for identifying outliers, they should be used in conjunction with other methods to ensure a comprehensive analysis of the data, such as establishing the applicability domain (AD) of the models, even though that is more problematic for a large data set, which is not the case here. Nevertheless, for future work, the AD of the derived models can easily be established using the confidence estimation approach proposed by Roy and co-workers (Ref. [4]).
Response 5: We performed extended analysis of outliers, results added to the text with discussion and figures 4-5, highlighted in green color.
Comments 6: Results for nanotoxicity multi-class classification problem – Figure 5 should include a footnote explaining the meaning of the plot ordinates to help the reader. Indeed, as it is right now, it is difficult to associate with the former classes given in Table 5 of section 3.4. At least, the authors should add the numbering in such Table 5. Figure 6 is also very difficult to inspect, and should be improved.
Response 6: Figure 5 is corrected with better visual understanding of toxicity classes. Figure 6 is prepared and inserted in high quality, zooming in allow to see the graphics for all seven classes in detail. As an alternative, we can put Figure 6 as separate 14 figures in supplementary materials, if this will suit better readability.
Comments 7: Applying trained ML models to other nanoparticles – Are the poor prediction obtained for large concentrations of external nanoparticles explainable? Or is just a too small data set? Please comment.
Response 7: Such poor prediction for large concentrations cannot be caused by bio determinants, therefore, we assume that the model needs to be retrained on new additional data.
Comments 8: Potential limitations: This section is just a resume of the work that was performed! No discussion about potential improvements in the modelling tools or about the applicability of the employed approach for future nanoparticles toxicity endeavours are really given and these would be beneficial.
Response 8: Potential improvements to trained models come from adding new training data obtained from new types of nanoparticles. Good predictive results were obtained for nanoparticles that belong to the group of transition metals of the fourth period. We see great potential of applicability of the employed approach to predict the toxicity of microelements from others groups after additional training on new data.
Comments 9: Additional suggestion: It is highly recommended that the authors follow FAIR principles by depositing the proposed approach (code, models, data set) in a relevant and publicly accessible repository. Doing so will allow other researchers to access, understand, and potentially build upon their work.
Response 9: The full dataset presented in this article is not readily available because the data are part of an ongoing study. Regarding the trained models, we have made them publicly available, the models can be tested on sample datasets: link
Comments 10: Finally, the authors should carefully revise the whole manuscript to correct typos and other aspects, such as adding references regarding the “Optuna” software, the “plot_tree” method (page 10, line 16) and feature importance analysis (page 10, line 322) used.
Response 10: References and footnotes added to the text, highlighted in green color.
Reviewer 2 Report
Comments and Suggestions for Authors
Authors developed models on the set of three types of nano particles on the basis of experimental measurements performed on three types of nano particles. One of them (SiO2) shows the viability, which is completely different from other two. It is difficult to believe that such model can present good predictions for nano particles of different type like Mn2O3 or Co3O4). The ‘good’ predictions presente3d in section 4.3 should be discussed more details.
Explain the scale in figure 4b (100 – 200 %) and figure 4d (0 – 200 %).
Author Response
Comments 1: Authors developed models on the set of three types of nano particles on the basis of experimental measurements performed on three types of nano particles. One of them (SiO2) shows the viability, which is completely different from other two. Explain the scale in figure 4b (100 – 200 %) and figure 4d (0 – 200 %).
Response 1: Silicon oxide, unlike metals, stimulates the glow of the strain (metabolic functions), therefore the values do not decrease from 100 at a non-toxic level to 0 at a toxic level (metals), but on the contrary, they grow from 100% in the control and above to 200% in the experiment, this is probably due to the nature of the element and the possibility of its incorporation into cellular structures in molecular, not ionic form.
Comments 2: It is difficult to believe that such model can present good predictions for nanoparticles of different type like Mn2O3 or Co3O4). The ‘good’ predictions presented in section 4.3 should be discussed more details.
Response 2: We added normalized confusion matrices for prediction on two additional nanoparticles (see Fig. 8 and Fig. 9). The initial laboratory data for Mn2O3 nanoparticles do not contain records corresponding to the Tox and NTOX2+ classes. The model shows fairly good prediction results for the EC80 class. The initial laboratory data for Co3O4 nanoparticles do not contain records corresponding to the NTOX2+ class. The model shows fairly good prediction results for the NTOX class.
Reviewer 3 Report
Comments and Suggestions for Authors
This paper summarizes the current status of nanoparticle toxicity research and predicts the toxicity of three nanoparticles by using a regression model and a multiclass classification model. The algorithm is described in detail and the model has achieved good results. However, there is a lack of innovation and some necessary experiments. Thus, I suggest a major revision with following comments:
1. First, in the Introduction, the purpose of describing recent research results should be to express the outstanding results or method improvements of your study.
2. In the Problem statement, the assignment and mathematical formulation of each parameter are described in detail. Is there any original calculation method for machine learning models?
3. Is it possible to reveal more detail of each experiment in your study? For example, in a toxicity research model of the single nanoparticle, I'm not very clear that if you detect the characteristic value of a nanoparticle in the overall toxicity study of multiple nanoparticles, or additional toxicity experiment using independent nanoparticle?
4. In addition to the luminescence of the bacteria, adding other methods, such as flow cytometry, to indicate the activity state of bacteria can make the experiment more rigorous.
5. The result in Figure 4 is very well, but adding additional test sets can make the overall result more rigorous.
6. In the Result of 4.3, more groups should be added in each concentration of each nanoparticle
7. Regarding dynamic toxicity prediction, adding cell or animal experiments will greatly increase the persuasiveness of the entire model.
Author Response
Comments 1: First, in the Introduction, the purpose of describing recent research results should be to express the outstanding results or method improvements of your study.
Response 1: Most relevant studies only consider the problem of binary classification of toxicity (toxic/nontoxic) or quantitative prediction of toxicity. We trained a regression model for predicting the toxicity of a nanoparticles depending on their concentration in the nutrient solution at a fixed point in time of the stationary growth phase. We trained a multi-class classification model for predicting the toxicity class (7 classes) of nanoparticles depending on their concentration in the nutrient solution at a fixed point in time of the stationary growth phase.
Comments 2: In the Problem statement, the assignment and mathematical formulation of each parameter are described in detail. Is there any original calculation method for machine learning models?
Response 2: In our study, we used standard methods for assessing the quality of machine learning models.
Comments 3: Is it possible to reveal more detail of each experiment in your study? For example, in a toxicity research model of the single nanoparticle, I'm not very clear that if you detect the characteristic value of a nanoparticle in the overall toxicity study of multiple nanoparticles, or additional toxicity experiment using independent nanoparticle?
Response 3: For each type of nanoparticles, an assessment of the suspension was performed, which is an integral indicator at a given concentration. The size and zeta potential of nanoparticles were calculated based on three measurements as an average.
Comments 4: In addition to the luminescence of the bacteria, adding other methods, such as flow cytometry, to indicate the activity state of bacteria can make the experiment more rigorous.
Response 4: We agree with this comment, additional methods for determining viability would improve the accuracy of the models, but we will not have studies based on the flow cytometry method due to the lack of the necessary equipment.
Comments 5: The result in Figure 4 is very well, but adding additional test sets can make the overall result more rigorous.
Response 5: We agree with this comment, additional test dataset would improve the accuracy of the models, but the data obtained so far are part of an ongoing study. We are planning to conduct more experiments during next year.
Comments 6: In the Result of 4.3, more groups should be added in each concentration of each nanoparticle
Response 6: Any threshold of concentrations from EC10 to EC90 can be identified, but usually listed seven groups are given, since they correspond to toxic, semi- toxic and sub-inhibitory doses.
Comments 7: Regarding dynamic toxicity prediction, adding cell or animal experiments will greatly increase the persuasiveness of the entire model.
Response 7: This study is focused on building a database of toxic effects on prokaryotes. Experiments on eukaryotic cells and animals are planned for next year to test sub inhibitory doses.
Round 2
Reviewer 3 Report
Comments and Suggestions for Authors
The author's reply and revisions effectively addressed my questions. Overall, the project demonstrates certain innovations and enhances the content related to the nanoparticle toxicity experiment. I look forward to seeing the implementation of this toxicity prediction method in cellular and animal studies in the future.
Author Response
Comments 1: "The author's reply and revisions effectively addressed my questions. Overall, the project demonstrates certain innovations and enhances the content related to the nanoparticle toxicity experiment. I look forward to seeing the implementation of this toxicity prediction method in cellular and animal studies in the future."
Response 1: Thank you for your comment. Implementation of proposed toxicity prediction models in cellular and animal studies is planned for next year.